# Position: Medical Large Language Model Benchmarks Should Prioritize Construct Validity

**Ahmed Alaa** [1,2]   **Thomas Hartvigsen** [3]   **Niloufar Golchini** [1]   **Shiladitya Dutta** [1]   **Frances Dean** [1,2]
**Inioluwa Deborah Raji** [1]   **Travis Zack** [2]

## Abstract

Medical large language models (LLMs) research often makes bold claims, from encoding clinical knowledge to reasoning like a physician. These claims are usually backed by evaluation on competitive benchmarks—a tradition inherited from mainstream machine learning. But how do we separate real progress from a leaderboard flex? Medical LLM benchmarks, much like those in other fields, are arbitrarily constructed using medical licensing exam questions. For these benchmarks to truly measure progress, they must accurately capture the real-world tasks they aim to represent. In this position paper, **we argue that medical LLM benchmarks should—and indeed can—be empirically evaluated for their construct validity**. In the psychological testing literature, "construct validity" refers to the ability of a test to measure an underlying "construct", that is the actual conceptual target of evaluation. By drawing an analogy between LLM benchmarks and psychological tests, we explain how frameworks from this field can provide empirical foundations for validating benchmarks. To put these ideas into practice, we use real-world clinical data in proof-of-concept experiments to evaluate popular medical LLM benchmarks and report significant gaps in their construct validity. Finally, we outline a vision for a new ecosystem of medical LLM evaluation centered around the creation of valid benchmarks.

## 1. Introduction

In recent years, medical Large Language Models (LLMs) have garnered significant attention, with a growing body of research examining their capabilities. These range from en-

[1]UC Berkeley [2]UCSF [3]University of Virginia. Correspondence to: Ahmed Alaa <amalaa@berkeley.edu>.

*Proceedings of the 42nd International Conference on Machine Learning*, Vancouver, Canada. PMLR 267, 2025. Copyright 2025 by the author(s).

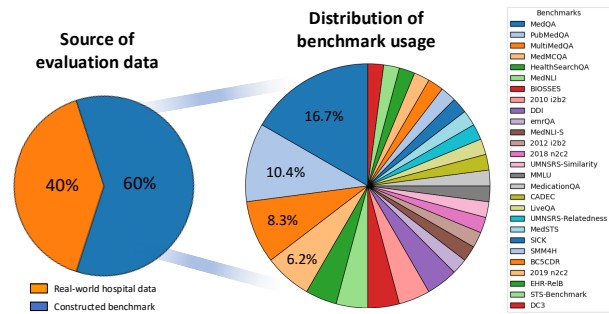

*Figure 1.* **Overview of evaluation datasets for medical LLMs.** We analyzed the evaluation datasets used in the 100 most cited papers on medical LLMs over the past 5 years. The majority (60%) of studies assess models on public benchmarks constructed based on medical exams, while 40% rely on (private or public access) real-world hospital data. There is no clear consensus on a standard benchmark—though MedQA is the most frequently used.

coding clinical knowledge (Singhal et al., 2023) to making differential diagnoses (McDuff et al., 2023), summarizing complex medical texts (Van Veen et al., 2024), mimicking clinical reasoning (Savage et al., 2024; Brodeur et al., 2024), and even demonstrating empathy in patient interactions (Maida et al., 2024). Yet the question of how to evaluate these capabilities remains a subject of ongoing debate. In the world of medicine, the gold standard for generating evidence is the randomized controlled trial (RCT). While some studies indeed conduct RCTs with meaningful real-world outcomes (Li et al., 2023; Brodeur et al., 2024), most research on medical LLMs leans on competitive benchmarks—an evaluation practice inherited from the broader machine learning community (Donoho, 2024; Orr & Kang, 2024).

Benchmarks have always played a central role in driving progress in machine learning, and we do not dispute their importance. Indeed, progress in areas such as computer vision over the past decade would likely not have occurred without benchmarks like ImageNet (Russakovsky et al., 2015). Public benchmarks are indispensable tools for the community to enable frictionless democratization of progress (Donoho, 2024; Recht, 2024). However, the landscape of medical LLM benchmarks remains fluid and fragmented. While flagship medical LLMs, such as Med-PaLM 2 (Singhal et al.,

2025), tout progress by evaluating on a select set of popular benchmarks, much of existing research uses custom, one-off evaluation datasets, and the field has yet to converge on consensus benchmarks for medical capabilities (**Fig. 1**).

**We need medical LLM benchmarks to gauge progress,** yet existing options leave much to be desired. As in other machine learning domains, most benchmarks proposed in the literature are arbitrarily constructed. In the context of medical LLMs, the tendency to anthropomorphize these models has led researchers to adopt human medical examinations—such as the United States Medical Licensing Examination (USMLE)—as evaluation benchmarks (Pal et al., 2024). Yet, as any clinician can attest, real-world clinical practice bears little resemblance to these exams (Raji et al., 2025). Unlike benchmarks in tasks like ImageNet classification, where performance claims are relatively inconsequential, medical LLM benchmarks carry significant weight, as they implicitly suggest general clinical skills. To ensure such claims are valid, our benchmarks must be as faithful to the complexities of real-world clinical practice as possible.

The USMLE, like other professional licensing exams such as the bar for lawyers, is designed to evaluate baseline requisite knowledge prior to being given the agency to practice medicine. These exams act as proxies for human readiness, much in the same way that the SAT exam is only a proxy for the success of a student in college. Over decades, they have been refined through observations of human performance and an understanding of cognitive abilities thought to underpin professional competence (Haist et al., 2013). However, there is no compelling reason to assume these exams serve the same purpose for LLMs as they do for humans. This points to a deeper issue in the current discourse surrounding LLM evaluation: the lack of a principled framework for determining what constitutes a good benchmark and how benchmark performance relates to real-world utility.

In this position paper, **we argue that medical LLM benchmark datasets should be deliberately designed and quantitatively evaluated to ensure *construct validity*.** Furthermore, in the context of medicine, this quantitative evaluation can uniquely leverage real-world data, such as electronic health records (EHR). Construct validity refers to the *degree to which a test or measurement accurately represents the concept or construct it is intended to measure.* This concept was first introduced by Cronbach & Meehl (1955) in the context of discussions on the validity of psychological tests. In this paper, we draw an analogy between LLM benchmarks and such psychological tests, and explain how methods for quantitative assessment of the validity of psychological tests can be applied to medical LLM benchmarks using EHR data. **We envision a new standard for evaluating medical LLMs:** each benchmark should be associated with explicit claims and empirically evaluated by hospital systems to ensure that its construct validity supports those claims.

Our position is motivated by the observation that **LLM capabilities, much like psychological traits, are viewed as latent "constructs" that lack operational definitions.** These models are no longer regarded as mere statistical predictors performing narrowly defined classification tasks on structured inputs and outputs, but as "agents" with emergent "capabilities" in open-ended tasks (Wei et al., 2022). Just as the multi-faceted and complex constructs of intelligence or depression cannot be measured in the same way as temperature is measured with a thermometer, the ability of LLMs to reason cannot be assessed in the same way we evaluate image classifiers. For our evaluation to support such a claim, we must move beyond practices where benchmarks are accepted based solely on their subjective face validity.

Psychometrics has long tackled this challenge that machine learning is only now beginning to face: how to design tests that accurately measure latent constructs (Strauss & Smith, 2009). **We propose that machine learning take a cue from psychology and develop a new science of benchmarking, focused on creating principled tools to evaluate the construct validity of its benchmark datasets.** At its core, construct validity asks whether a test truly captures the real-world phenomenon it claims to measure. This notion is particularly apt for medical benchmarks, as medicine generates a wealth of real-world data from clinical practice (Evans, 2016), which makes it an ideal starting point for building a benchmarking science grounded in empirical reality.

**Much of the research critiquing popular medical LLM benchmarks or proposing new ones is ultimately an effort to enhance construct validity.** For example, Johri et al. (2025) introduce an evaluation framework based on simulated patient-clinician conversations rather than the clinical vignettes used in medical exams. The motivation is that traditional question-answering benchmarks fail to capture the nuances of real-world patient-clinician interactions, where skills like comprehensive history-taking and open-ended questioning are crucial. This is fundamentally a question of construct validity—i.e., to what extent does a benchmark measure the skills required for real-world diagnostic reasoning? **In this paper, we equip researchers with a natural framework for discussing, evaluating and constructing medical LLM benchmarks using general principles that transcend specific tasks**, instead of addressing limitations in existing benchmarks through ad-hoc solutions.

Discussions of benchmark validity are conspicuously absent from mainstream machine learning literature. While some work has critiqued the validity of image classification benchmarks—Raji et al. (2021) qualitatively analyzed the construct validity of ImageNet and GLUE, while Fang et al. (2024) empirically tested whether ImageNet progress translates to real-world datasets—construct validity has yet to

become a standard consideration in benchmark design, including for LLMs. Empirical evaluation of construct validity would clarify what benchmark performance actually signifies and help make sense of discrepancies in model rankings across benchmarks in model leaderboards (Pal et al., 2024).

In the rest of this paper, we elaborate on our position. In Section 2, we draw a parallel between psychological tests and LLM benchmarks, and explain how LLM capabilities mirror psychological constructs and what this means for their evaluation. Section 3 provides an overview of validity theory and the various notions of construct validity, using the example of a depression test to illustrate how these concepts can be applied to medical LLM benchmarks. In Section 4, we examine how real-world clinical data can be used to empirically assess the validity of popular medical LLM benchmarks and highlight validity gaps in current benchmark datasets. Finally, we outline our vision for an evaluation ecosystem for medical LLMs that prioritizes valid benchmarks and discuss alternative views to model evaluation.

## 2. LLM Benchmark Datasets as Analogues to Psychological Tests

LLM benchmark datasets and psychological tests share a key conceptual similarity: **both aim to measure a "latent construct"—an abstract, unobservable trait that is not operationally defined**. In psychology, constructs such as intelligence, depression, or working memory cannot be directly measured like height or temperature; instead, they are assessed through standardized tests designed to capture behaviors or responses indicative of the underlying trait. For example, an IQ test does not measure intelligence itself but evaluates problem-solving, pattern recognition, and reasoning skills, which are taken as proxies for intelligence. Similarly, LLM benchmarks do not directly measure its ability to "reason" or "understand" but instead assess performance on tasks assumed to reflect these capabilities, such as answering medical questions or summarizing text. Psychological testing generally involves five key components:

1. **Test Subject:** The individual being evaluated.
2. **Latent Construct:** The psychological construct that the test measures. This might include personality traits, cognitive abilities, or mental health conditions.
3. **Test Instrument:** A set of items (questions, tasks, or stimuli) presented to the test subject to elicit responses.
4. **Test Score:** A quantitative measure derived from the subject's responses to the items in the test instruments.
5. **Inference on Test Result:** The interpretation of the test score in relation to the underlying latent construct.

In our analogy, different LLMs (e.g., GPT-4, PaLM and Claude) serve as test subjects, their capabilities (e.g., mathematical reasoning) as latent constructs, evaluation on benchmark datasets (e.g., GSM8K, MATH (Cobbe et al., 2021)) as

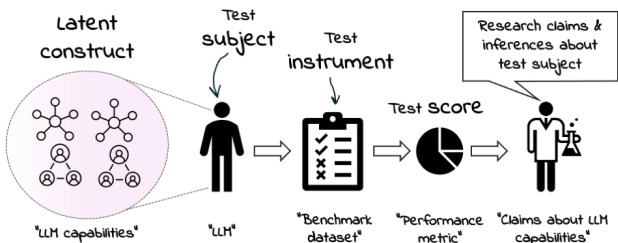

*Figure 2.* **Analogy between LLM benchmarks and psychological testing.** Tests aim to evaluate latent constructs that are theoretically conceived but not directly observable. The validity of a test depends on how well inferences drawn from its scores align with the underlying construct being measured across test subjects.

test instruments, performance metrics on benchmarks as the test scores, and researchers' claims about model capabilities as inferences based on the resulting test scores (**Fig. 2**).

The validity of psychological tests and the risks of their misinterpretation have long been debated in psychometrics. In *The Mismeasure of Man*, Stephen J. Gould criticized the validity of craniometry and IQ tests as measures of intelligence, highlighting how they have been misused to justify politicized views of biological determinism (Gould, 1996). Similarly, our research community must critically examine whether we are mismeasuring models through benchmarks with limited (construct) validity, and whether our research claims risk enabling the misuse of LLMs in critical applications such as medicine. The analogy between LLM benchmarks and psychological tests extends beyond medicine, but in the next section, we take a closer look at how psychologists have tackled test validity and how insights from validity theory can help us empirically assess the relevance of medical LLM benchmarks as measures of progress.

## 3. Validity Theory and the Rethinking of Medical LLM Benchmarks

The concept of *construct validity* was first introduced in a seminal paper by Cronbach & Meehl (1955), which categorized test validity into three distinct types: *criterion*, *content*, and *construct*—together forming the classical view on validity of tests. Later, Samuel Messick (Messick, 1998) argued that construct validity subsumes all three, establishing it as the overarching framework in modern validity theory. In this section, we outline both the classical (Section 3.1) and modern (Section 3.2) perspectives on validity theory and explore their relevance to medical LLM benchmarks.

### 3.1. Classical view: A tripartite theory of validity

Validity theory provides a systematic framework for determining whether a psychological test truly measures the psychological construct of interest—for instance, whether the Big Five Personality Test captures personality traits (Barrick

& Mount, 1991) or the Stanford-Binet Intelligence Scale assesses intelligence (Laurent et al., 1992). Under the classical view, validity of a test can be evaluated through three distinct methods, which we detail below. As a running example, consider the *Beck Depression Inventory* (BDI), a psychometric test developed in the 1960s by Beck et al. (1961) to assess depression severity through a multiple-choice questionnaire (Jackson-Koku, 2016). BDI has been the subject of extensive research on its validity as a measure of the "depression construct" (Richter et al., 1998). We show how each type of validity applies to BDI and, by extension, how these principles can inform the validation of medical LLM benchmarks.

> **1) Criterion validity** examines *how well a test predicts or correlates with an outcome that represents successful expression of the construct being measured.*

A criterion in validity theory is an external, observable outcome or measure that we have a reason to believe the construct should predict. For instance, a criterion for validating the BDI could be a clinical diagnosis of depression made by a psychiatrist using DSM-5 criteria. Criterion validity comes in two forms: *predictive* or *concurrent* (Barrett et al., 1981). Predictive validity is assessed when the test results precede the criterion measure, while concurrent validity is evaluated when both are measured within a similar timeframe (Guion & Cranny, 1982). A test demonstrates criterion validity if its scores meaningfully correlate with the chosen criterion.

**Example of criterion validation for BDI tests.** Concurrent validity of the BDI can be assessed by examining whether BDI scores correlate with other established measures of depression. For example, (Ambrosini et al., 1991) evaluated the concurrent validity of BDI scores in outpatient adolescents by comparing them to diagnoses generated by the Kiddie-Schedule for Affective Disorders and Schizophrenia (K-SADS) and a 17-item clinician-rated depression scale derived from the K-SADS (Puig-Antich & Ryan, 1986). Predictive validity can be assesed by examining if BDI scores predict future depressive episodes or treatment outcomes. For instance, (Green et al., 2015) assessed the predictive validity of BDI by examining the association of its suicide item with future suicide attempts or deaths by suicide.

**Criterion validation of medical LLM benchmarks.** Suppose we focus on *diagnostic reasoning* as the LLM capability of interest and use a benchmark like MedQA, which comprises clinical vignettes paired with multiple choice diagnosis questions, to test the diagnostic reasoning construct. A strong criterion for validating this benchmark would be the diagnostic accuracy of an LLM on real-world patient cases. In other words, if the score achieved by the medical LLM on MedQA predicts its accuracy in diagnosing actual patients, then the benchmark demonstrates *criterion (predictive) validity*. Fortunately, empirically assessing this is feasi-

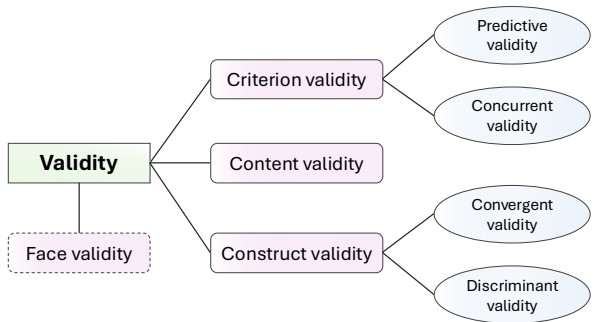

*Figure 3.* **Outline of different types of test validity evidence.** Beyond subjective face validity, the classical tripartite theory categorizes validity into criterion, content, and construct validity. Modern perspectives view construct validity as the overarching concept, with face, predictive, concurrent, convergent, and discriminant validity serving as distinct sources of evidence for construct validity.

ble—one could match clinical vignettes with corresponding patient charts and ICD codes in EHRs and test whether the medical LLM accuracy on MedQA vignettes correlates with its performance on similar real-world patient cases.

Criterion validation of medical LLM benchmarks resemble procedures already used to assess medical licensing exams by evaluating their correlation with future clinical and professional performance. For instance, Swanson et al. (1996) studied knowledge retention in fourth-year medical students by determining whether their performance on medical exam items predicted their scores on similar items in the future. Similarly, Norcini et al. (2014) studied the correlation between USMLE scores and long-term professional success.

> **2) Content validity** is established by demonstrating that the *test items are representative of the content domain of interest.* Typically, content validity is established deductively, by first defining a domain and then systematically selecting items from it to form the test.

Criterion validity alone may be insufficient to establish the validity of a test. For example, while the BDI might successfully predict depression-related outcomes, it would fall short as a true measure of depression if it overemphasized symptoms like sleep disturbances and appetite changes while neglecting other aspects such as low mood and anhedonia. A valid test must capture the full breadth of the construct it aims to measure. As Cronbach & Meehl (1955) describe, the "universe" or "content domain" of a construct encompasses all possible ways to assess its various facets—for depression, this includes every conceivable way of evaluating mood, anhedonia, sleep disturbances, etc. In this sense, content validity assesses whether a test meaningfully represents this content domain rather than arbitrarily sampling from it.

**Example of content validation for BDI tests.** Content val-

idation of BDI can be performed by defining the "universe" of all possible depression symptoms based on DSM/ICD criteria and having an expert panel assess how well the items within BDI represent these symptoms. For example, to highlight the strengths and gaps in its coverage, Moran (1982) found that the BDI effectively captures 6 of the 9 DSM-III criteria for depression. However, two criteria—sleep disturbances and changes in eating behavior—are only partially addressed, while one, agitation, is entirely absent.

**Content validation of medical LLM benchmarks.** The content domain of medicine is exceptionally well-structured in comprehensive ontologies, taxonomies and tasks. Clinical ontologies (SNOMED, LOINC, RxNorm, ICD) provide granular classification of diseases, symptoms, procedures, and medications, while educational frameworks systematically map learning objectives in medical exams. These frameworks—notably Bloom's taxonomy for learning objectives and Miller's pyramid for clinical competence progression—offer ready-made structures for evaluating the coverage of clinical concepts, skills and tasks for medical LLM benchmark (Bloom et al., 1956; Miller, 1990).

Once the content domain is defined, content validation can be carried out by empirically evaluating how comprehensively a benchmark like MedQA or PubMedQA cover medical concepts relevant to a given patient population or health system. The relevance of individual concepts can be gauged by assessing their prevalence in real-world EHR data for the population of interest. Additionally, benchmark coverage for different types of patients can be evaluated since EHRs keep track of the demographic makeup of the population.

> **3) Construct validity** is involved whenever a test measures an attribute that is not *operationally defined*. In essence, it seeks to answer the question: *"What constructs account for variance in test performance?"*

Construct validity evaluates whether a test truly measures the *theoretical* construct it claims to capture. A test has construct validity if variations in scores across individuals can be primarily attributed to differences in that construct. This is a broader concept than criterion and content validity and was introduced by Cronbach & Meehl (1955) following the American Psychological Association Committee on Psychological Tests (1950–1954) (APA, 1954), which concluded the limitations of relying solely on criterion and content validation to establish the validity of tests. Today, as we discuss in Section 3.2, "construct validity" has evolved into an umbrella term that encompasses all aspects of test validity.

Returning to the BDI example, depression is not "operationally defined" because it cannot be measured directly like temperature with a thermometer. Instead, depression is a theoretical construct that must be assessed indirectly through its observable manifestations. Establishing the construct valid-

ity of BDI requires examining how its scores relate to other variables that are theoretically associated with depression. Construct validity is typically classified into: *convergent validity*, which indicates that test scores correlate with other measures of the same construct, and *discriminant validity*, which ensures that test scores do not correlate with measures of unrelated constructs (Campbell & Fiske, 1959). The primary goal of construct validity is to identify the theoretical constructs that explain variance in test performance. For the BDI, this means ensuring that score differences primarily reflect variations in levels of depression among individuals.

**Example of construct validation for BDI.** Schotte et al. (1997) investigated the construct validity of BDI in a large sample of unipolar depressive inpatients. The study used factor analysis to demonstrate that BDI captures two distinct dimensions of the depression construct in the subjects' responses, psychological/cognitive and somatic/vegetative, and that these dimensions correlate with external measures based on Dexamethasone suppression biomarkers.

**Construct validation of medical LLM benchmarks.** A medical LLM benchmark has construct validity if it reliably distinguishes between models based on their proficiency in the clinical skill being evaluated. Simply put, models that perform well in the evaluated task should score higher, while weaker models should score lower. For example, consider diagnostic reasoning: the construct validity of MedQA as a test of this skill can be assessed by comparing model rankings on MedQA with their rankings in real-world diagnostic accuracy evaluated on real-world patient cases. If MedQA is a valid measure of diagnostic reasoning, its rankings should generalize to real-world clinical cases—but they would not necessarily predict performance on unrelated tasks, such as medical text summarization or treatment planning. In this light, the fact that models rank differently across benchmarks that supposedly evaluate the same task—such as those on the Open Medical-LLM Leaderboard—raises important questions about the construct validity of these benchmarks.

### 3.2. Modern view: "All validity is construct validity"

Contemporary perspectives on test validity no longer treat "types of validity" as distinct categories under the tripartite model. While Cronbach & Meehl (1955) originally defined construct validity as a separate type alongside content and criterion validity, Messick (1989) argued that all forms of validity evidence ultimately contribute to the interpretation and use of constructs.[1] In his seminal 1989 work, Messick reframed what were once considered separate validity types as different forms of evidence supporting construct validity. This unified view emphasized that validation is not about

---

[1]Thirty years after his 1955 paper, Cronbach himself aligned with Messick's view, agreeing that the classical theory is ultimately about supporting construct interpretations (Cronbach, 1989).

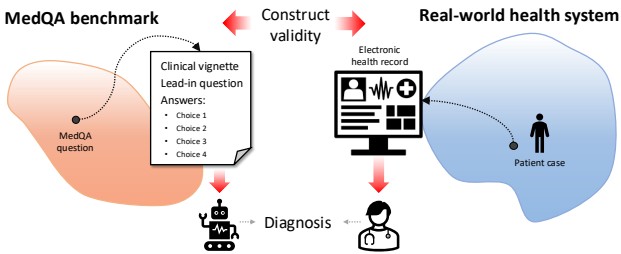

*Figure 4.* **Empirical validation of the MedQA benchmark using EHR data.** Each MedQA item consists of a clinical vignette, a question, and multiple-choice answers, while each EHR patient case includes a clinical note and a corresponding clinical decision. To assess the validity MedQA, we empirically test whether strong performance on benchmark items reflects the ability of an LLM to encode and apply medical knowledge in real-world practice.

the test itself but about the interpretations and applications of test scores, with all validation efforts aimed at supporting inferences about constructs. That is, even ostensibly straightforward processes, such as criterion prediction or content sampling, are grounded in theoretical assumptions about constructs. The idea that "all validity is construct validity" was later reinforced and expanded through the influential work of Michael Kane (Kane, 2012; 2001). The Educational Testing Service (ETS), which administers standardized tests such as TOEFL and GRE, currently adopts this unified view in its test design manuals (Kane & Bridgeman, 2017).

## 4. Empirical Validation of Medical LLM Benchmarks using EHR Data

Validity theory provides a theoretical framework for assessing whether medical LLM benchmarks reflect real-world tasks. However, for benchmark validation to become a standard practice, we need clear, practical validation procedures. We argue that benchmark validation is a distinct research agenda, and medicine provides an ideal testing ground due to the vast availability of real-world clinical data from EHRs which can enable direct comparisons between benchmarks and the real-world tasks they are meant to represent (Adler-Milstein et al., 2015; Blumenthal & Tavenner, 2010).

In this section, we demonstrate that practical validation of medical LLM benchmarks is feasible, even as the rigorous strategies for empirical validation remain an open question. As a proof of concept, we evaluate the MedQA benchmark (Jin et al., 2021) using real-world EHR data from the University of California, San Francisco medical center. MedQA was constructed from USMLE questions to evaluate the medical reasoning and knowledge of LLMs in a multiple-choice format. Each item in the benchmark comprises a clinical vignette, lead-in question and answer choices (**Fig. 4**). Full experimental details are provided in the Appendix.

The MedQA benchmark is meant to evaluate the **construct**

**of medical reasoning and knowledge**—strong performance is often taken as evidence that an LLM *"encodes medical knowledge."* However, for this claim to hold, a model that excels on MedQA should also demonstrate the ability to reason and apply relevant medical knowledge in real-world clinical practice. To demonstrate the feasibility of empirical evaluation of benchmark validity, we introduce proof-of-concept validation procedures, which we structure within the classical validity framework outlined in Section 3.1.

### 4.1. Criterion validity of MedQA

We evaluate the criterion validity of MedQA by **testing whether the accuracy of an LLM on its multiple-choice questions predicts its performance in real-world clinical decisions requiring the same medical knowledge**. To conduct this experiment, we matched each MedQA question to 10 patient cases in the EHR where the clinical decision relied on the same medical knowledge being tested. Specifically, we extracted the correct answer for each question in MedQA and categorized it as either a drug or a diagnosis. Drugs were mapped to RxNorm codes, while diagnoses were mapped to their respective SNOMED codes. Using these standardized codes, we queried clinical notes in EHR data to retrieve 10 physician progress notes in which the drug or diagnosis appeared in the "Assessment and Plan" section. This process yielded a dataset of benchmark items, each paired with 10 real-world patient encounters where the tested medical knowledge was applied (**Fig. 4**).

|  | MedQA | Real-world data | |
|---|---|---|---|
|  | *Accuracy* | *Accuracy* | *α* |
| Llama 3 | 0.54 | 0.48 | 0.56 |
| GPT-4 | 0.71 | 0.28* | 0.29 |
| Chimera Llama | 0.60 | 0.45 | 0.48 |
| Biomerge | 0.57 | 0.36 | 0.49 |
| Orpomed | 0.49 | 0.24 | 0.38 |
| JSL MedLlama | 0.61 | 0.37 | 0.49 |
| PMY MedLLama | 0.75 | 0.36 | 0.45 |

*Table 1.* Predictive validity of MedQA. (*GPT-4 refrained from answering 57% of real-world questions; unanswered cases were counted as an error. Accuracy on answered cases was 0.64.)

We evaluated state-of-the-art LLMs: GPT-4 and Llama 3.0, as well as the leading models from the Medical-LLM leaderboard: Chimera Llama, Biomerge, JSL Medllama, PMY Medllama, Orpomed, and OpenBioLLMC (Pal et al., 2024). For each model, we evaluate accuracy on MedQA, accuracy on real-world data, as well as the following criterion:

$$\alpha = P(\textbf{Correct on real-world case} \mid \textbf{Correct on MedQA}).$$

The metric $\alpha$ is the conditional accuracy of an LLM on real-world cases that resemble the clinical vignettes it correctly answered in the benchmark. This metric serves as a measure of predictive validity—if correctly answering a benchmark

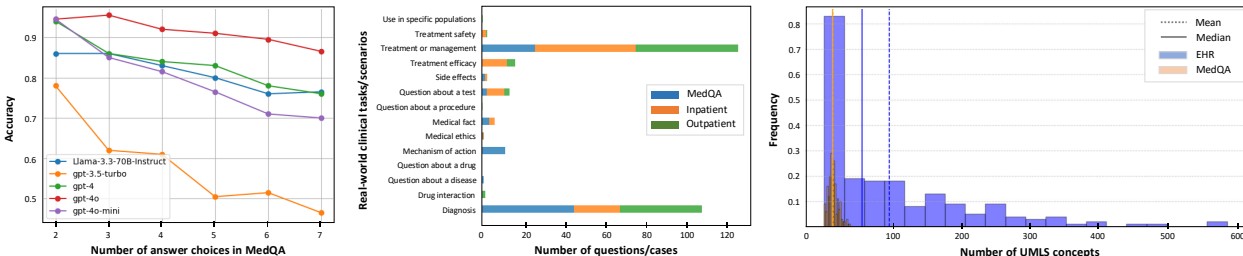

*Figure 5.* **Left:** Impact of number of answer choices in MedQA on accuracy. **Middle:** Distribution of clinical tasks and scenarios in MedQA and real-world data. **Right:** Distribution of the number of UMLS concepts in MedQA vignettes and real-world clinical notes.

question implies a high probability of accuracy on similar real-world cases, then the benchmark effectively predicts real-world performance. To estimate $\alpha$, we filter MedQA to retain only questions the LLM answered correctly and then evaluate its accuracy on the matched real-world cases.

The results in **Table 1** show a modest correlation between LLM performance on the MedQA benchmark and real-world cases. In a scenario of perfect criterion validity, we would expect $\alpha$ to approach 1. However, our experiment show that $\alpha$ offers little improvement over the overall accuracy of an LLM on real-world data, suggesting that correctly answering benchmark questions does not strongly predict real-world clinical performance on diagnostic and treatment decisions similar to those in the benchmark. Additionally, we observe a general drop in accuracy when models transition from the benchmark to real-world cases. A likely explanation is the contrived format of MedQA, which provides answer choices that are absent in real-world clinical reasoning. Notably, as the number of answer choices in MedQA increases, performance drops across all models (**Fig. 5**).

There are several reasons why a benchmark might show poor criterion validity. For one, the idealized patient presentations in clinical vignettes may differ significantly from the messy, real-world clinical notes that are often cluttered with irrelevant information that could make it hard for model performance on the benchmark to generalize. Another possibility is that matching patients based on the correct answer in MedQA might not represent the best measure of clinical similarity. To set a standard for criterion validity, researchers and clinicians could collaborate to define a more robust standard for comparing items in a benchmark and real-world patient cases in terms of their "clinical similarity."

### 4.2. Content validity of MedQA

To evaluate **how well MedQA represents real-world clinical concepts and contexts where medical knowledge is applied**, we designed an experiment to strip away surface-level details and represent both MedQA questions and real-world clinical notes in a unified content format. We leveraged the Unified Medical Language System (UMLS) (Bodenreider, 2004) to define the "content domain" of medical knowledge

(Bodenreider, 2004). UMLS is a comprehensive ontology that integrates concepts from medical vocabularies such as SNOMED, MeSH, LOINC, RxNorm, and ICD. By comparing the UMLS concept coverage between MedQA questions and real-world clinical notes, we can determine how well MedQA represents the content domain of clinical practice. For concept extraction, we used cTAKES, a widely adopted NLP system in health informatics (Savova et al., 2010).

In addition to extracting UMLS concepts, we also examined how MedQA questions align with real-world clinical scenarios where medical knowledge is retrieved and applied across both inpatient and outpatient progress notes. To do this, we categorized each MedQA question and clinical note into one of 15 possible clinical scenarios, such as diagnosis, treatment management, and reasoning about treatment safety and side effects. This allows us to define the content domain via *(patient, task)* tuples—linking the patient cases that resemble MedQA vignettes and the real-world decision-making contexts that resemble MedQA questions (**Fig. 4**).

The results in **Fig. 5** indicate that while MedQA covers a range of real-world clinical scenarios, it disproportionately favors diagnostic questions over treatment-related ones. Additionally, real-world patient cases are associated with a significantly larger number of USMLE concepts compared to clinical vignettes in MedQA, which are intentionally streamlined to include only the information necessary to answer the question. In contrast, real-world medicine is messy—patient records often contain a flood of details, many of which are irrelevant to the immediate diagnostic or treatment task. This stark difference in complexity suggests that MedQA represents a much simpler content domain than real-world practice, which could explain its poor predictive validity in **Table 1**, since prior research has shown that LLMs are prone to distraction (Shi et al., 2023; Hager et al., 2024).

### 4.3. Construct validity of MedQA

The construct validity of MedQA holds if **variations in accuracy scores across LLMs reflect their actual ability to encode medical knowledge**. A straightforward way to assess this is by comparing model rankings between the benchmark and a real-world evaluation task. If MedQA genuinely

measures medical knowledge, the rankings of LLMs on the benchmark should align with how these models perform when retrieving and applying medical knowledge in real-world cases. However, as we see in Table 1, this is not the case: GPT-4 tops the rankings on MedQA, yet Llama 3 outperforms on real-world clinical notes. Interestingly, GPT-4 had a notably high non-response rate on real-world notes, which makes the interpretation of benchmark peformance even harder. Is MedQA truly measuring clinical knowledge or spurious statistical pattern matching?

Factor analysis is commonly used to assess the construct validity of psychological tests by examining whether test items correlate with real-world measures that reflect the intended psychological construct (Kang, 2013). Similarly, the construct validity of medical benchmarks can be evaluated by comparing model performance on categorized benchmark tasks with real-world clinical tasks in the same categories. Another approach is to prompt models with chain-of-thought reasoning and assess if their reasoning align with expert clinicians. If a model reaches the correct answer with flawed reasoning, it may indicate that the benchmark captures statistical patterns rather than medical understanding.

## 5. A Benchmark-Validation-First Approach to Medical LLM Evaluation

In the previous section, we showed that empirically evaluating the validity of medical LLM benchmarks is not only possible but well within reach. This opens the door for a new line of multidisciplinary research on developing methods for a new evaluation paradigm that prioritizes benchmark validation before model evaluation, rather than the other way around. In this envisioned evaluation practice, benchmarks are not arbitrary datasets but structured tests with a clearly defined construct, content domain, and validity criteria.

Our envisioned benchmark is not a foreign concept in clinical research—if anything, it aligns more closely with the structure of clinical studies, which evaluate specific interventions (constructs) within a defined population (content) against measurable outcomes (criteria). Yet, current medical LLM benchmarks fall short of this standard—we rely on ad hoc datasets, loosely defined metrics, and populations that lack clear relevance. Even our evaluation metrics often fail to capture what truly matters. For instance, Goodman et al. (2024) highlights how standard accuracy measures like ROUGE fail to reflect the quality of LLM-generated clinical summaries. MedQA, for its part, does not specify a particular disease, population, or demographic focus. And when it comes to constructs, none of the existing benchmarks in **Fig. 1** are designed with a well-defined theoretical model of LLM capabilities in mind. By incorporating construct validity principles, we can not only evaluate the relevance of existing benchmarks but also develop new valid ones.

One might ask: why concern ourselves with the construct validity of toy benchmarks when we could simply evaluate LLMs on real-world data? While some recent benchmarks, such as MedHELM[2], incorporate real-world data and tasks, even those benchmarks involve design choices that must be scrutinized for construct validity. For instance, the selection of which real-world tasks to include necessarily reflects assumptions about what constitutes competent performance. A medical benchmark might emphasize diagnostic accuracy over patient communication skills, which are difficult to simulate in silico, or prioritize rare disease identification over common primary care scenarios. Similarly, the method of adaptation from real-world contexts to evaluation formats, such as converting clinical decision-making into multiple-choice questions or summarizing complex patient interactions into standardized prompts, introduces its own biases and potential misalignment with real-world practice. Even the choice of success metrics, whether focusing on exact match accuracy, clinical safety, or downstream patient outcomes, implicitly makes assumptions about what medical LLM competence should look like. Thus, while real-world data provides valuable grounding, it does not eliminate the fundamental challenge of construct validity; it merely shifts the locus of concern from artificial tasks to the inevitable abstractions required to make real-world complexity amenable to systematic evaluation.

More broadly, many hospitals are often reluctant to share data due to the risk of violating privacy regulations. In our envisioned paradigm, hospitals would not need to share raw data at all. Instead, they could serve as benchmark validators by locally evaluating public benchmarks and reporting validation scores to researchers. Similar to model leaderboards, we could have leaderboards for evaluation benchmarks, where the most thoroughly validated benchmarks rise to the top. Model leaderboards could prioritize evaluation on these top-ranked benchmarks. Such framework creates a competitive ecosystem where researchers vie to develop benchmarks validated by the most hospitals, while developers select benchmarks endorsed by health systems most relevant to their deployment setting.

## 6. Alternative Views

In this position paper, we argued for a fundamental rethinking of how we evaluate medical LLMs, particularly as their capabilities grow more abstract and their deployment contexts become increasingly open-ended. Since benchmarks are not created equal, we need tools and a shared framework for evaluating and comparing them in terms of how well they reflect real-world tasks in clinical practice. We proposed leveraging the tools used to evaluate the construct validity of psychological and educational tests to establish an

---

[2]https://crfm.stanford.edu/helm/medhelm/latest/

empirical science for medical benchmark evaluation.

There are **alternative views** on the future of medical LLM evaluation, all of which challenge the very concept of benchmarking itself. One perspective argues that, given the evolving nature of healthcare—changes in populations, the introduction of new drugs, etc (Finlayson et al., 2021)—we cannot rely on an the same benchmark to evaluate progress over time. A static benchmark may create an illusion of progress, but it will inevitably lose relevance as medicine advances. Another viewpoint stresses that evaluation should prioritize clinical utility, specifically the impact of LLMs on clinical decision-making (Hager et al., 2024), rather than focusing on benchmark performance—an argument that has also been made in the context of diagnostic tests (Bossuyt et al., 2012). A third view posits that, as LLMs are seen as agents, their evaluation should use conversational simulators rather than static datasets (Mehandru et al., 2024; Johri et al., 2025).

While these alternative approaches may play a role in the future evaluation of medical LLMs, we argue that benchmarks will likely remain the most frictionless means of reproducing and evaluating new LLMs. Evaluating clinical utility typically requires models to be embedded within a health system, which is not feasible to most researchers, while adaptive evaluations or agent-based simulations are essentially just other forms of constructed tests. Crucially, the requirement for construct validity applies to any evaluation instrument, whether a static benchmark, a simulated environment, or an interactive conversational system. Even if the field moves toward evaluation methods beyond benchmarks, the core issue of construct validity will continue to be relevant.

## Acknowledgments

This work was supported in part by a Marcus Program in Precision Medicine award from the George and Judy Marcus Innovation Fund and a National Science Foundation Graduate Research Fellowship under Grant No. DGE 2146752.

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

## A. Figure 1. Overview of Evaluation Datasets for Medical LLMs

We conducted a systematic PubMed search to identify open-access articles evaluating large language models (LLMs) in clinical contexts, focusing on publications from the last five years. Using the PubMed query function, we retrieved PubMed IDs (PMIDs) and obtained abstracts and full texts via the PubMed Central Open Access (PMCOA) API in structured BioC JSON format for further analysis. Our search yielded 361 papers, representing publicly available clinical LLM research in PubMed at the time. To facilitate manual review, we selected the **top 100 most-cited papers** and extracted key metadata from their abstracts and methods sections. Each paper was categorized based on two criteria:

1. **Source of Evaluation Data**

   - **Real-World Hospital Data**: Papers that used clinical datasets from electronic health records (EHRs), patient charts, or other real-world medical sources (e.g., MIMIC-III, institutional hospital records).
   - **Constructed Benchmark**: Papers that evaluated LLMs using synthetic or researcher-designed data, such as expert-generated clinical vignettes, hypothetical case studies, or simulated patient interactions.

2. **Benchmark Used for Evaluation**

   - **Public Benchmark**: If the paper used a publicly available dataset for LLM evaluation (e.g., PubMedQA, MedQA, MultiMedQA, MMLU, i2b2, n2c2, BC5CDR, or other established biomedical NLP benchmarks), the specific benchmark was recorded in the dataset.
   - **Constructed Dataset**: If the paper used a dataset specifically created for the study (e.g., handcrafted clinical scenarios or institution-specific data not publicly available), the benchmark field was left blank.

## B. Table 1. Predictive Validity of the MedQA Benchmark

To evaluate different models on their ability to handle clinical scenarios, we matched MedQA questions to real word clinical notes in a systematic process. We started with the complete Multiple Choice Questions (MCQs) from the Step 1 and Step 2 sections of the MedQA dataset. Each MCQ was divided into two parts: (1) the prompt, which included the context of the question, and (2) the extracted question, which represented the specific inquiry about the prompt.

From the answer choices provided for each question, we isolated the correct answer and assigned it to one of two categories: either a drug or a diagnosis. Correct answers identified as drugs were mapped to their corresponding RxNorm codes, while diagnoses were matched to their respective SNOMED codes. Using these standardized codes, we queried the clinical notes to retrieve approximately 10 notes containing either the drug or the diagnosis within the "Assessment and Plan" section of a physician's progress note.

The retrieved clinical notes included all sections typically structured in the SOAP (Subjective, Objective, Assessment, and Plan) format. For our analysis, we excluded the "Plan" section to avoid bias and paired the remaining note content with the extracted question and its answer choices, forming a dataset that resembled standard MCQs with real-world clinical context.

This curated dataset was then used to evaluate top-performing models, including GPT-4, GPT-3.5, and Llama 3.0, as well as leading models from Hugging Face's MedQA leaderboard: Chimera Llama, Biomerge, Orpomed, JSL Medllama, PMY Medllama, and OpenBioLLMC.

## C. Figure 5. Content Validity Experiments

### C.1. Left: Impact of Number of Answer Chouces in MedQA on Accuracy

We test across 3 different axes: model, dataset, and perturbation. For the models, we tested on GPT-3.5-Turbo, GPT-4, GPT-4o, GPT-4o-mini, and Llama-3.3-70b. We test across 3 different datasets: MedQA (4 option version), MedMCQA, and MMLU (specifically the "anatomy", "clinical knowledge", "college medicine", "college biology", "medical genetics", and"professional medicine" subsets). For MedQA and MMLU, we draw from the test split and for MedMCQ we draw from the dev split.

### C.2. Middle: Task Coverage Comparison between MedQA and Real World Data Methodology

We extracted a sample of 100 inpatient progress notes and 100 outpatient progress notes to represent real-world medical documentation. For the real-world clinical notes, we isolated the Assessment and Plan sections, as these contain the

physician's primary clinical reasoning and decision-making processes. The extracted sections were then input into the model for categorization. We applied the following standardized prompt across all three datasets (MedQA, inpatient notes, and outpatient notes) to classify the primary task being performed in each note:

*"Identify which of the following categories best characterizes the primary task performed by the physician in this note:"*

1. Diagnosis

2. Medical fact

3. General question about a drug

4. General question about a disease

5. Treatment or management

6. Drug interaction

7. Side effects

8. Question about a procedure

9. Question about a test or measurement

10. Mechanism of action

11. Definition

12. Treatment efficacy

13. Treatment safety

14. Use in specific populations

15. Medical ethics

The model was instructed to respond strictly in the following format: **[Number: Category]** (e.g., **"1. Diagnosis"**).

### C.3. Right: UMLS Concept Methodology

To analyze the distribution of the UMLS concept of medical claims within the MedQA benchmark, we systematically processed all questions (n = 1,273) in the data set. First, we iterated through each question and saved it as an individual TXT file to facilitate batch processing. These TXT files were then processed using the cTAKES (clinical Text Analysis and Knowledge Extraction System) API.

The cTAKES pipeline categorized the identified UMLS concepts into predefined semantic types, including diseases, laboratory tests, medications, procedures, and signs or symptoms. For each question, we link the extracted concepts back to their corresponding MedQA question, ensuring a clear mapping between the question and its associated UMLS concepts.

Each UMLS Concept Unique Identifier (CUI) served as a unique key, allowing us to quantify the total number of concepts extracted across the entire dataset. By aggregating and analyzing these CUIs, we characterized the distribution of medical concepts present in the MedQA benchmark, providing insights into the diversity and scope of clinical knowledge tested by the data set.

C.3.1. CONCEPT DISTRIBUTION PER SECTION METHODOLOGY

To analyze the concept distribtion across structured sections of an Electronic Health Record (EHR), we collaborated with a medical doctor to identify the key sections commonly used by physicians to evaluate clinical scenarios. These sections served as a framework for categorizing information within the USMLE questions in the MedQA dataset.

Using this framework, we systematically input each USMLE question into a standardized structure and prompted GPT-4o with the following instruction:

```
Please categorize the inputted questions into the following structured JSON format
by only extracting information from the input and not adding anything additional:
{
    "Patient History": {
        "Demographics": "...",
        "Chief Complaint": "...",
        "Patient Presentation": "...",
        "Past Medical History": "...",
        "Current Medications": "...",
        "Review of Symptoms": "..."
    },
    "Objective": {
        "Vital Signs": "...",
        "Physical Exam": "...",
        "Laboratory Findings": "...",
        "Radiographic Findings": "...",
        "Pathology Findings": "..."
    },
    "Question": "..."
}
Question:
{q['question']}
```

This prompt ensured that each question was categorized into predefined EHR sections, extracting only the information present in the question text without adding external details. After categorizing all questions, we applied the cTAKES pipeline to each section of the structured questions to identify and map UMLS concepts.

For each section (e.g., *Patient History*, *Objective*), the cTAKES analysis categorized the extracted concepts into semantic types such as diseases, laboratory tests, medications, procedures, and signs or symptoms. The UMLS Concept Unique Identifiers (CUIs) were linked back to their respective sections and questions, enabling a granular analysis of concept distributions within the MedQA benchmark.

