# OpenReview forum: "Position: Medical Large Language Model Benchmarks Should Prioritize Construct Validity"
_ICML.cc/2025/Position_Paper_Track — ICML 2025 Position Paper Track oral_

### Official Review · Reviewer_p6pz · 2025-03-03

**Significance:** 4
**Argument Clarity:** 2
**Rating:** 4
**Confidence:** 5

**Questions:**

Please see the strength and weakness section.

**Discussion Potential:**

3

**Paper Summary:**

The central argument of the paper is that medical LLM benchmarks must be empirically validated for construct validity to ensure they meaningfully measure real-world clinical capabilities. The authors assert that relying on traditional medical exam-based benchmarks (e.g., USMLE-based datasets) may lead to misleading conclusions about model competence. By applying psychometric validity frameworks and leveraging real-world clinical data, the paper calls for a paradigm shift in medical AI evaluation towards scientifically grounded benchmark development.

**Position:**

Yes

**Position In Title:**

Yes

**Related Work:**

2

**Strengths And Weaknesses:**

## Strengths

1. The paper presents a critical perspective on the evaluation benchmarks for medical large language models (LLMs), advocating for rigorous assessment of **construct validity**—a psychometric concept that evaluates whether a test genuinely measures the intended construct. As an LLM researcher, I find this notion novel and refreshing, offering a compelling lens through which to evaluate LLM performance.

2. The authors propose a framework for assessing the construct validity of medical LLM benchmarks, drawing on principles from validity theory in psychology. This framework introduces a systematic and theoretically grounded approach to LLM evaluation, making it a valuable contribution to the field.

3. The paper highlights critical gaps in benchmark validity, demonstrating that high performance on MedQA does not necessarily translate to strong real-world clinical performance. While this finding is not entirely unexpected, it serves as an important empirical contribution to ongoing discussions within the LLM research community.

---

## Weaknesses

1. **Limited Scope of Empirical Validation**
   The empirical evaluation is largely confined to MedQA, raising concerns about the generalisability of the findings to other medical benchmarks. As the authors acknowledge, there are over 20 state-of-the-art (SOTA) medical benchmarks, and newly released models typically report performance on 8–10 of the most widely used ones. Given that different benchmarks assess distinct aspects of medical knowledge, it is unsurprising that MedQA alone may not generalise to real-world electronic health record (EHR) challenges. However, what about other benchmarks? Expanding the analysis to 5–10 additional benchmarks would provide a broader perspective. A simple linear regression analysis could further examine whether a **performance scaling law** exists between medical benchmark scores and real-world clinical tasks. For instance, benchmarking 10 models against 10 benchmarks and treating these as input features while using a real-world benchmark score as the outcome could yield insights. If the **R²** of the regression is low, it would indicate that a linear combination of existing benchmark scores fails to predict real-world performance.

2. **Lack of Empirical Validation for the Proposed ‘Construct Validity’ Framework**
   While the construct validity framework is conceptually appealing, its practical implementation remains vague, particularly for LLM researchers unfamiliar with psychometric principles. To improve clarity, the authors should present a concrete case study comparing a **construct validity-driven evaluation system** with a conventional benchmark-based system. This would help illustrate its practical utility and effectiveness.

3. **Insufficient Discussion of Existing Medical Benchmarks**
   The paper does not sufficiently acknowledge that some existing benchmarks already evaluate LLMs on real-world clinical tasks. Notable examples include benchmarks based on **MIMIC-III datasets** [1][2][3]. These works provide direct evaluations of LLMs in realistic medical contexts and should be discussed more extensively to ensure the paper’s claims are well-supported.

4. **Lack of Technical Details**
   Several aspects of the study require additional technical clarity. For example, the authors should provide more details on the **real-world dataset** used in Table 1, the criteria used to define the **‘100 most cited papers’** in Figure 1, and other methodological choices. Each figure and dataset should be accompanied by sufficient explanations to ensure reproducibility and transparency.

---

## Conclusion

The paper presents a conceptually compelling argument for rethinking medical LLM evaluation through the lens of construct validity. However, its claims lack **sufficient empirical support**, particularly in terms of broader benchmark validation and case study demonstrations. Strengthening the empirical validation, incorporating more benchmarks, and addressing existing real-world evaluations would enhance the paper’s impact and credibility.

**Support:**

2

---

> ### Author Rebuttal · Authors · 2025-03-31
>
> We thank the reviewer for their valuable feedback, comments, and suggestions. Many of the points raised relate to empirical evaluations of construct validity. As a general clarification, in alignment with the requirements of the position paper track, we emphasize that our paper is meant to convey a position rather than delivering exhaustive empirical analysis of existing benchmarks. The empirical results included in the paper are meant to provide evidential support for our main argument and show the feasibility of empirical evaluation of construct validity. We acknowledge that this paper does not (and is not intended to) exhaustively explore all possible findings, tools, or methods within this emerging paradigm.
>
> Below, we provide a detailed point-by-point response to the limitations raised by reviewer.
>
> **1- Limited Scope of Empirical Validation**
>
> While it is true that our empirical evaluation focuses on MedQA, we do not believe this raises concerns about generalizability. Our central claim **is not that all benchmarks lack construct validity**, but rather that *failing to consider construct validity can lead the community to collectively adopt benchmarks that are poorly aligned with real-world tasks*, as we argue has occurred with MedQA. By showing that the **single most popular benchmark** (MedQA) fails to meet basic construct validity standards, we provide empirical support for our broader argument.
>
> As the reviewer notes, many recently released models report performance across multiple benchmarks. As we discuss in lines 111–114, these models often rank differently depending on the benchmark as in the Open Medical-LLM leaderboard. While many benchmarks use a similar MCQ format and are intended to assess the same capabilities, discrepancies in model rankings highlight the need for a deeper evaluation framework. This is precisely where construct validity plays a critical role: it offers a principled way to interpret these discrepancies and assess which benchmarks are most meaningful. Importantly, we do not believe that MedQA fails to generalize simply because it captures a narrow slice of medical knowledge that could be supplemented by other benchmarks. Rather, our experiment  demonstrates that MedQA is not representative of the types of clinical decisions in real-world settings.
>
> That said, we appreciate the reviewer’s suggestion for including more datasets to explore a potential performance scaling law between medical benchmark scores and real-world clinical tasks. We have already begun replicating our analysis using two additional benchmarks (MedMCQA and medical-MMLU) and plan to include several models. We aim to incorporate this expanded analysis into the camera-ready version of the paper.
>
> **2- Lack of Empirical Validation for the ‘Construct Validity’ Framework**
>
> We recognize that many LLM researchers may be unfamiliar with the literature on validity theory, and that’s why we dedicated Sections 2 and 3 to outlining its foundational concepts, and we believe we already provided concrete proof-of-concept evaluation in Section 4. To reiterate, the goal of our position paper is not to introduce a specific eval method for immediate use, but to argue that the field needs to transform from evaluating LLMs using tools designed for statistical models to tools more akin to those used in the social sciences. Just as psychometrics required decades of research to develop methods for validating psychological tests, we hope our paper will help catalyze a similar research agenda around evaluating the validity of LLM benchmarks. Ultimately, comparing construct validity-driven benchmarks to conventional benchmarks will require years of work, including user studies and RCTs, which fall well beyond the scope of a position paper.
>
> **3- Insufficient Discussion of Existing Medical Benchmarks**
>
> We thank the reviewer for highlighting this. While MIMIC-III has been widely used to benchmark traditional ML models (based on biomarkers and vital signs), we are not aware of commonly used LLM benchmarks based on this data. The references [1–3] mentioned were not provided, but we will include a discussion in Sec. 5 of the camera-ready version that addresses real-world datasets such as MIMIC and MedHELM.
>
> **4- Lack of Technical Details**
>
> Thank you for pointing this out. The real-world dataset used in our study comes from an academic medical center, which we have not named in order to preserve anonymity during the review process. We have prepared a comprehensive Appendix with methods behind Fig. 1, Table 1, and Fig. 5. In accordance with rebuttal guidelines, we are unable to share external links at this stage, but we would be happy to address any technical questions during the author-reviewer discussion period. The full Appendix will be included in the camera-ready version, and we also plan to release the code for reproducing all MedQA-related experiments.

---

> > ### Comment · Reviewer_p6pz · 2025-04-02
> >
> > Thank the authors for their response. I am generally happy about the rebuttal, except for point 1. Providing the additional results before the rebuttal ends would give me sufficient reason to raise my score.
> >
> > Reference missing previously:
> > [1] Bardhan, Jayetri, et al. "Drugehrqa: A question answering dataset on structured and unstructured electronic health records for medicine related queries." arXiv preprint arXiv:2205.01290 (2022).
> >
> > [2] Lehman, Eric. "Learning to Ask Like a Physician: a Discharge Summary Clinical Questions (DiSCQ) Dataset."
> >
> > [3] Kweon, Sunjun, et al. "Ehrnoteqa: A patient-specific question answering benchmark for evaluating large language models in clinical settings." Preprint (2024).

---

> > > ### Author Response · Authors · 2025-04-07
> > >
> > > Thank you for your response and for providing detailed references. We will incorporate all three references into the final version of the paper.
> > >
> > > Following your suggestion, we replicated the analysis presented in Figure 5 of our submission using two additional widely-used benchmarks: MedMCQA and MMLU. As expected, the results from these analyses demonstrate consistent trends in content, criterion, and construct validity across all benchmarks examined. The new results can be accessed via this [anonymous link](https://drive.google.com/file/d/1hDvMRdJ8lEAdXyTILvpVGxNm0Ipx6USn/view?usp=sharing). Given the limited time remaining in the rebuttal period, we welcome suggestions from the reviewer regarding any additional benchmarks that might provide conceptually valuable insights for inclusion in the camera-ready version of the paper.
> > >
> > > We hope this fully addresses your concerns, and we would be happy to discuss any further comments during the remainder of the rebuttal period.

---

### Official Review · Reviewer_vpet · 2025-03-06

**Significance:** 4
**Argument Clarity:** 4
**Rating:** 5
**Confidence:** 4

**Questions:**

1. Please consider and respond to the points raised in the weaknesses section. Otherwise, great work!

**Discussion Potential:**

4

**Paper Summary:**

This work argues that the design of benchmarks for medical large language models should prioritize construct validity. In other words, they should be carefully designed such that performance on the benchmark coincides with the unobserved construct that the evaluation aims to target. The work further argues that designing benchmarks in this way is feasible and practical. They support this with a proof-of-concept study using MedQA (i.e. a set of questions derived from the US Medical Licensing Exam) paired with data extracted from an electronic health record system. Criterion, content, and construct validity are assessed through (1) comparison of the relationship between performance on comparable cases between the two domains, (2) coverage of relevant concepts and tasks in MedQA, and (3) consistency of ranking of performance of different models across the two domains. Ultimately, the work argues for grounding benchmark design in terms of its correspondence with performance in concrete clinical contexts.

## Update after rebuttal
No changes after the rebuttal. I recommend that the paper be accepted.

**Position:**

Yes

**Position In Title:**

Yes

**Related Work:**

4

**Strengths And Weaknesses:**

# Strengths
* In my view, this is a strong submission to the position paper track.
* The position is clearly stated, with direct arguments that are supported by evidence (both primary evidence generated as a part of the study and more broadly from the literature).
* I believe that there is significant potential for this work to meaningfully shift practice in the design of benchmarks for medical large language models.
* The presentation of the work as a whole is very clear. I found the background material regarding validity theory to be particularly well-presented with the appropriate level of detail and rigor for a machine learning audience.
* The experiments are convincing as a proof-of-concept to demonstrate the general feasibility of designing benchmarks to incorporate construct validity considerations.
* I think this work will prompt productive discussion from the community. In my experience, there has been increasing interest from the community in recent years in “measurement”, which includes construct validity as a central principle. As examples, “Measurement and Fairness” (Jacobs and Wallach 2021) has been influential in shifting practice in evaluations of fairness across several applied fields; the recent workshop “Evaluating evaluations” at Neurips 2024 broadly emphasized measurement considerations; and there has further been several recent/contemporaneous works that argue for shifting away from MedQA-style evaluations towards those that are more meaningful clinically (see for example Raji et al 2025, Fleming et al 2023, Shah et al 2025). I bring these works up primarily to emphasize the broad interest in the questions that this work aims to address. The work does meaningfully advance on those prior works through its grounding in formal measurement theory and by providing an empirical proof-of-concept to demonstrate feasibility. In my view, this is a meaningful advance because it demonstrates that construct validity may be empirically tested and is not just an abstract design principle (as one potential critique of the overarching agenda).

# Weaknesses
* The work implicitly makes the assumption that the intended use of medical large language models is for clinical use cases. This comes through in several places in the manuscript through discussion of the importance of grounding in clinical practice, the use of EHR data, and in positioning hospitals as potential arbitrators of benchmark validity. As a counter to this, I would argue that medical knowledge in large language models is also relevant for use cases that are not necessarily clinical. For example, if we are interested in consumer health use cases or retrieval and summarization of medically-relevant content by a search engine. I would recommend that the authors clarify the scope of their argument. This is a relatively minor critique.
* The scale of the empirical evaluation is relatively small and preliminary. However, I think that the scope is reasonable for a position paper.

# References
1. Jacobs, Abigail Z., and Hanna Wallach. "Measurement and fairness." Proceedings of the 2021 ACM conference on fairness, accountability, and transparency. 2021.
2. Raji, Inioluwa Deborah, Roxana Daneshjou, and Emily Alsentzer. "It’s Time to Bench the Medical Exam Benchmark." NEJM AI 2.2 (2025): AIe2401235.
3. Fleming, Scott L., et al. "Medalign: A clinician-generated dataset for instruction following with electronic medical records." Proceedings of the AAAI Conference on Artificial Intelligence. Vol. 38. No. 20. 2024.
4. Shah N., Pfeffer, M., Liang, P.. “Holistic Evaluation of Large Language Models for Medical Applications”. https://hai.stanford.edu/news/holistic-evaluation-of-large-language-models-for-medical-applications

**Support:**

4

---

> ### Author Rebuttal · Authors · 2025-03-31
>
> We thank the reviewer for their valuable feedback and for highlighting the important contributions by Jacobs & Wallach, Raji et al., and others. We fully agree that, like these influential works, our paper aims to inspire the development of a new science of benchmark evaluation grounded in principles from measurement theory. We hope our work contributes to similarly productive conversations and advances in the field. Below are point-by-point responses to the weaknesses mentioned in the review.
>
>
> **1-** It is true that our paper primarily assumes LLMs are being developed for clinical use cases. This focus is motivated by the fact that much of the current medical LLM literature centers on claims about medical reasoning capabilities of LLMs in decision making contexts. However, as the reviewer rightly points out, LLMs with medical reasoning skills may have many other impactful applications beyond direct clinical decision-making. We will clarify the broader scope of LLM use cases in both the introduction and discussion sections of the camera-ready version.
> That said, we believe that construct validity remains highly relevant even for administrative and consumer health applications of LLMs. For example, in clinical text summarization, there is ongoing debate around what it truly means to perform well in real-world settings, and these are questions that are not adequately addressed by existing benchmarks. We refer the reviewer to a thoughtful commentary on this topic:
>
> *Goodman, K. et al., AI-Generated Clinical Summaries Require More Than Accuracy, JAMA, 2024.*
>
>
> **2-** The preliminary scope of our evaluation reflects our belief that a broader science of evaluation is still in its early stages, and we do not yet have all the answers for how it should be fully developed. Our experiments are intended as proof-of-concept demonstrations to show the feasibility of empirically evaluating benchmark validity, rather than as an exhaustive analysis of all existing benchmarks. Our focus in this paper is on articulating our position by introducing the machine learning community to foundational ideas from measurement theory and outlining an agenda for evaluating construct validity in LLM benchmarks.

---

> > ### Comment · Reviewer_vpet · 2025-04-02
> >
> > Thank you for the thoughtful response. These minor changes address my concerns. Best of luck with the submission!

---

### Official Review · Reviewer_rFKQ · 2025-03-14

**Significance:** 4
**Argument Clarity:** 4
**Rating:** 4
**Confidence:** 4

**Questions:**

See suggestions for improvement

**Discussion Potential:**

3

**Paper Summary:**

This position paper argues that current benchmarks for evaluating medical Large Language Models (LLMs), often derived from licensing exams like the USMLE, lack demonstrated relevance to real-world clinical practice. The authors advocate for prioritizing construct validity, a concept borrowed from psychometrics, which assesses whether a test truly measures the underlying capability it intends to evaluate.

# update after rebuttal

I am happy to recommend the acceptance of the paper if the suggestions could be incorporated into the revision or camera ready. Thanks.

**Position:**

Yes

**Position In Title:**

Yes

**Related Work:**

2

**Strengths And Weaknesses:**

Hope the feedback is valuable for the authors and helps improve the quality in the revision or camera ready. Thanks!

Pros:
1. Effectively identifies and articulates the significant issue of questionable real-world relevance in current medical LLM benchmarks.
2. Successfully introduces and explains the concept of construct validity from psychometrics, providing a robust theoretical foundation for evaluating benchmarks.
3. The comparison between LLM benchmarks and psychological tests is clear, insightful, and helps justify the need for validity testing.
4. Moves beyond critique to propose a concrete approach—empirical validation using real-world data—and outlines a vision for a more reliable evaluation ecosystem.

Suggestions for improvement:
1. The literature review could be more comprehensive. The authors also may consider referencing more recent medical benchmarks such as [1,2,3,4] in the revision.
2. While the paper strongly advocates for construct validity, the empirical demonstration in Section 4.3 is relatively brief. It primarily relies on rank correlation (already suggested by the criterion validity results in Table 1). Expanding this with even a preliminary application of other mentioned methods (like qualitative analysis of reasoning patterns or a simplified factor analysis proxy) would strengthen the proof-of-concept.
3. The process of matching MedQA items to EHR cases (lines 293-302) is crucial but described briefly. More detail on the matching criteria, potential ambiguities, and inter-rater reliability (if applicable) would increase confidence in the results. How sensitive are the findings to different matching strategies?
4. The paper operationalizes real-world performance based on accuracy related to specific drugs/diagnoses mentioned in notes. While practical, this is a narrow slice of clinical practice. Acknowledging the limitations of this proxy for broader "clinical knowledge application" would be beneficial.
5. The paper correctly identifies the need for a better measure of "clinical similarity" (lines 365-371) but doesn't offer concrete suggestions beyond collaboration. Briefly outlining potential dimensions (e.g., patient demographics, comorbidities, reasoning complexity) could make the call to action more tangible.


[1] FMBench: Benchmarking Fairness in Multimodal Large Language Models on Medical Tasks https://arxiv.org/abs/2410.01089

[2] ClinicalBench: Can LLMs Beat Traditional ML Models in Clinical Prediction? https://arxiv.org/abs/2411.06469

[3] CliBench: A Multifaceted and Multigranular Evaluation of Large Language Models for Clinical Decision Making https://arxiv.org/abs/2406.09923

[4] CLIMB: A Benchmark of Clinical Bias in Large Language Models https://arxiv.org/abs/2407.05250

**Support:**

3

---

> ### Author Rebuttal · Authors · 2025-03-31
>
> We thank the reviewers for their valuable feedback, which will greatly help us improve the revised and camera-ready version of the paper.
>
> **1-** Thanks for pointing out these additional references. We will add discussion of the corresponding benchmarks in the discussion section of the final paper.
>
> **2-** Thank you for this excellent suggestion. To analyze the clinical reasoning patterns of LLMs, we broke down each question in the MCQ benchmark into distinct sections (e.g., symptoms, chief complaint, etc.) and prompted the models being evaluated to identify which sections contributed to their answer, using this as a proxy for reasoning. In the revised or camera-ready version of the paper, we will include results from a factor analysis comparing these reasoning patterns between benchmark questions and real-world clinical data.
>
> **3-** We have prepared a comprehensive Appendix with the methods behind Figure 1, Table 1, and Figure 5, including our approach to patient matching, which is also described in Section 4.1. Patient matching can be done using various criteria, each would reflect different levels of granularity in patient contexts used to ascertain the matches. In our experiments, we match each benchmark question with 10 or more real-world patient cases where the diagnosis or prescription decision aligns with the correct answer from the benchmark. This decision-based matching focuses on whether the model selects the correct medication or diagnosis both in the benchmark and in actual clinical contexts; this is a minimal expectation if the benchmark is to reflect real-world utility.
>
> Our matching strategy takes a pragmatic, outcome-oriented approach by aligning benchmark items with real-world clinical decisions. Other matching strategies, such as matching based on clinical similarity of patient presentations as we mention in Lines 365-371, are also possible. That said, if a benchmark does not generalize even under coarse decision-based matching, it is unlikely to generalize under more granular criteria.
>
> We emphasize that the purpose of our experiment is to demonstrate the feasibility of empirically evaluating the construct validity of benchmarks, and the failure of some prominent benchmarks under rudimentary tests of validity. Determining the best methods for evaluating construct validity, and empirically evaluating the extent to which different benchmarks are valid, remains an open and important problem. We hope that our paper inspires further research on patient matching to develop more nuanced empirical methods for future research on benchmark validation.
>
> **4-** Thank you for pointing this out. We will acknowledge the broader scope of “clinical knowledge” beyond diagnostic and drug-related questions in the discussion section. That said, our focus on this narrower subset reflects the current focus of most medical LLM benchmarks on these types of questions. Within our framework, this narrow focus in current benchmarks represents a limitation in content validity, as it fails to capture the full range of clinical knowledge required in real-world practice.
>
> **5-** We will include additional suggestions in the discussion section, such as patient matching based on clinical note embeddings or itemized clinical context. Please note that our paper is intended to offer a conceptual framework for thinking about the validity of benchmarks, with the goal of inspiring future research to explore these questions in greater depth. We do not claim to have definitive answers regarding the best empirical tools for matching or validation, but rather aim to highlight the importance of developing such methods in future research.

---

### Decision · Program_Chairs · 2025-04-30

**Decision:**

Accept (oral)

**Comment:**

The paper propose to use construct validity from psychometrics for LLM evaluation in the medical domain. All reviewers are in consensus that this is a strong position paper.